# HMGB1 amplifies ILC2-induced type-2 inflammation and airway smooth muscle remodelling

Zhixuan Loh[1,2], Jennifer Simpson[1,3], Ashik Ullah[1,3], Vivian Zhang[3], Wan J. Gan[1], Jason P. Lynch[1,3], Rhiannon B. Werder[1,3], Al Amin Sikder[1,3], Katie Lane[1], Choon Boon Sim[4], Enzo Porrello[4], Stuart B. Mazzone[1,5], Peter D. Sly[6,7], Raymond J. Steptoe[8], Kirsten M. Spann[9], Maria B. Sukkar[10], John W. Upham[8], Simon Phipps[1,3,7] *

1 School of Biomedical Sciences, The University of Queensland, Queensland, Australia, 2 Institute for Molecular Bioscience, The University of Queensland, Queensland, Australia, 3 QIMR Berghofer Medical Research Institute, Queensland, Australia, 4 Murdoch Children's Research Institute, The Royal Children's Hospital, Melbourne, Australia, 5 Department of Anatomy and Neuroscience, University of Melbourne, Victoria, Australia, 6 Children's Health and Environment Program, Child Health Research Centre, University of Queensland, Queensland, Australia, 7 Australian Infectious Diseases Research Centre, The University of Queensland, Queensland, Australia, 8 UQ Diamantina Institute, The University of Queensland, Queensland, Australia, 9 School of Biomedical Sciences, Queensland University of Technology, Queensland, Australia, 10 Graduate School of Health, Faculty of Health, University of Technology Sydney, Ultimo, NSW, Australia

* simon.phipps@qimrberghofer.edu.au

**Data Availability Statement:** All relevant data are within the manuscript and its Supporting Information files.

## Abstract

Type-2 immunity elicits tissue repair and homeostasis, however dysregulated type-2 responses cause aberrant tissue remodelling, as observed in asthma. Severe respiratory viral infections in infancy predispose to later asthma, however, the processes that mediate tissue damage-induced type-2 inflammation and the origins of airway remodelling remain ill-defined. Here, using a preclinical mouse model of viral bronchiolitis, we find that increased epithelial and mesenchymal high-mobility group box 1 (HMGB1) expression is associated with increased numbers of IL-13-producing type-2 innate lymphoid cell (ILC2s) and the expansion of the airway smooth muscle (ASM) layer. Anti-HMGB1 ablated lung ILC2 numbers and ASM growth *in vivo*, and inhibited ILC2-mediated ASM cell proliferation in a co-culture model. Furthermore, we identified that HMGB1/RAGE (receptor for advanced glycation endproducts) signalling mediates an ILC2-intrinsic IL-13 auto-amplification loop. In summary, therapeutic targeting of the HMGB1/RAGE signalling axis may act as a novel asthma preventative by dampening ILC2-mediated type-2 inflammation and associated ASM remodelling.

## Author summary

Asthma can start at any time in life, although most often begins in early childhood. Wheezy viral bronchiolitis is a major independent risk factor for subsequent asthma. However, key knowledge gaps exist in relation to the sequelae of severe viral bronchiolitis

**Funding:** This work was supported by an NHMRC project grant (S.P. and J.W.U.; ID1023756), an equipment grant from the Rebecca L. Cooper Medical Research Foundation (S.P.), an Australian Infectious Disease Research Excellence Award (S. P.), and an Australian Research Council Future Fellowship (S.P.). The funders had no role in study design, data collection and analysis, decision to publish, or preparation of the manuscript.

**Competing interests:** The authors have declared that no competing interests exist.

and the pathogenic processes that promote type-2 inflammation and airway wall remodelling, cardinal features of asthma. Our study addresses this gap by identifying high-mobility group box 1 as a pathogenic cytokine that contributes to group 2 innate lymphoid cell-induced airway smooth muscle growth.

## Introduction

Asthma can start at any time in life, although most often begins in early childhood [1]. Wheezy viral bronchiolitis is a major independent risk factor for subsequent asthma [2], yet key knowledge gaps exist in relation to the pathogenic processes that link the two diseases. As the primary site of viral replication and a rich source of type-2 'instructive' mediators (e.g. IL-33), the airway epithelium is now recognised to be pivotal to disease inception through the local activation of type-2 innate lymphoid cells (ILC2) and mucosal dendritic cells. IL-33 is thought to be the main driver of ILC2 expansion and activation, however a growing body of evidence is beginning to implicate high-mobility group box 1 (HMGB1), another nuclear alarmin whose extracellular expression correlates with asthma severity [3–5]. In preclinical models of allergic asthma, anti-HMGB1 or genetic ablation of RAGE (receptor for advanced glycation endproducts), one of the receptors ligated by HMGB1, is protective [6–8]. Intriguingly, anti-HMGB1 or RAGE deficiency also diminishes IL-33-induced type-2 inflammation [6,9], suggesting that the HMGB1/RAGE axis contributes to a feed-forward circuit that amplifies type-2 inflammation. However, whether this pathway relates to ILC2 activation remains unknown.

One of the teleological roles of type-2 inflammation is to mediate wound repair [10,11]. For example, ILC2s can promote re-epithelialisation of the airway mucosa following viral infection or helminth infestation [12,13]. However, ILC2s can promote aberrant tissue repair, as observed in pulmonary fibrosis and liver fibrosis [14,15]. Curiously, it is not known whether ILC2s contribute to airway smooth muscle (ASM) remodelling, a cardinal feature of asthma and one of the main contributing factors to airway narrowing, loss of lung function, and airways hyperreactivity. Notably, structural alterations to the airway wall are evident in school-age children with asthma [16,17], and ASM area is increased in wheezy 'pre-asthmatic' children [18], highlighting the need for early intervention strategies to prevent airway remodelling, which is thought to be less plastic in later life [19].

Susceptibility to severe viral infections in infancy and the later development of asthma may stem from an impaired and/or delayed antiviral immune responses [20,21]. Indeed, a deficient interferon (IFN)-lambda response and greater epithelial sloughing to viral infections in children predicts later asthma persistence [22], thus linking impaired immunity to tissue damage. Downstream of toll-like receptor 7 and other virus-sensing pattern recognition receptors, IFN regulatory factor (IRF)7 regulates the antiviral IFN response to respiratory syncytial virus (RSV) and rhinovirus (RV) infection [23–25]. In a recent landmark paper comparing asthma risk in Amish (low risk) and Hutterite (4x higher risk) farm children [26], IRF7 was identified as a differentially expressed hub gene. In another clinical study, unbiased network analysis separated exacerbating children into IRF7^hi and IRF7^lo phenotypes, the latter associating with a deficient IFN response and shorter time to exacerbation recurrence [27]. Collectively, these findings suggest that impaired antiviral immunity predisposes to the inception of later asthma, however the pathogenic cellular and molecular processes that underpin this relationship remain ill-defined.

In this study, to determine whether HMGB1 is a contributing factor we used a preclinical model of neonatal viral bronchiolitis. We found that IRF7 deficiency has a profound effect on

the airway epithelium: namely increased viral burden, nuclear-to-cytoplasmic translocation of HMGB1, followed by epithelial sloughing into the lumen. The subsequent release of HMGB1 has a profound effect on the number and activation of lung ILC2s, which we show to promote ASM cell proliferation in an ILC2-ASM co-culture system. Although both ILC2s and ASM cells produce HMGB1, we show that ILC2-derived HMGB1 acts in an autocrine manner to amplify type-2 cytokine production. Together, our findings uncover an important role for HMGB1 in ILC2 activation and implicate ILC2s as important effectors of both type-2 inflammation and ASM thickening in early-life, predisposing to later asthma.

## Results

### Severe viral bronchiolitis induces epithelial damage and alarmin release

To assess the effect of impaired antiviral immunity on the development of severe viral bronchiolitis, we inoculated WT ($^{+/+}$) and IRF7-deficient ($^{-/-}$) neonatal (7 day old) mice with pneumonia virus of mice (PVM). As PVM is a mouse-specific pathogen (same genus as respiratory syncytial virus; RSV) it replicates efficiently in mice (unlike hRSV) and allows for the temporal study of host-pathogen interactions [28,29]. Compared to WT neonatal mice, viral load was massively elevated and clearance delayed in IRF7$^{-/-}$ neonatal mice (Fig 1A). This was associated with diminished production of antiviral cytokines, IFN-α (Fig 1B) and IL-12p40 (S1A Fig). As occurs clinically, the severe bronchiolitis that developed in IRF7$^{-/-}$ neonatal mice was associated with greater neutrophilic inflammation, tissue oedema, and pronounced sloughing of the airway epithelium (Fig 1C–1E). This epithelial damage was associated with elevated IL-33 and HMGB1 levels in the bronchoalveolar lavage fluid (BALF) and lungs of IRF7$^{-/-}$ neonatal mice (Fig 1F and 1G and S1B Fig). In contrast, other known inducers of type-2 inflammation, such as IL-25 and TSLP, were below the limit of detection. Mirroring the pattern of viral load in the epithelium, nuclear to cytoplasmic translocation of HMGB1 peaked at 7 dpi and remained elevated at 10 dpi (Fig 1H). Although HMGB1 levels in BALF persisted at 14 dpi, epithelial cell cytoplasmic HMGB1 had waned, suggesting that extracellular HMGB1 was derived from sloughed epithelial cells or another cellular source at this time. The functional activity of HMGB1 is affected by its oxidative state; all-thiol HMGB1 acts as a chemokine, whereas disulphide HMGB1 acts as a cytokine through the activation of MD-2/TLR4 [30]. Using Western blot to interrogate the isoforms, we observed that both all-thiol and disulphide HMGB1 were significantly elevated in the lung homogenate of infected IRF7$^{-/-}$ mice compared to their WT counterparts (Fig 1I). Thus, impaired antiviral immunity predisposed to airway epithelial damage, IL-33 release, and a sustained period of elevated HMGB1 expression.

### Severe viral bronchiolitis induces ILC2 activation and ASM proliferation

IL-33 and HMGB1 can both promote type-2 inflammation. Consistent with this, we found that BALF levels of IL-13 and IL-5, but not IL-4 (undetectable) and airway eosinophil numbers (Fig 2A and S1C Fig) were significantly elevated at 10 dpi in IRF7$^{-/-}$ compared to WT mice. This coincided with a marked increase in the numbers of 'natural' (Lin-, CD90+, CD45+, CD25+, ST2+, Sca-1+) ILC2s, which increased sharply between 7 and 10 dpi, and persisted until at least 14 dpi in the IRF7$^{-/-}$ mice (Fig 2B and S1D Fig). Consistent with the absence of IL-25, 'inflammatory' (Lin$^{-}$, CD90$^{+}$, CD45$^{+}$, KLRG1$^{high}$, Sca-1$^{-}$) ILC2s [31] were undetectable in the lung. To assess whether the natural ILC2s were the major source of IL-13, we crossed the IRF7$^{-/-}$ mice onto IL-4/IL-13 (4C13R) dual-reporter mice [31] to allow for *in situ* determination of cytokine production. Examination at the peak of the IL-13 response (10 dpi) revealed that in the IRF7$^{-/-}$ mice, >50% of the ILC2s were positive for dsRed (reporting for IL-13 expression), whereas in the WT infected mice ~5% of ILC2s expressed dsRed (Fig 2C). At 10

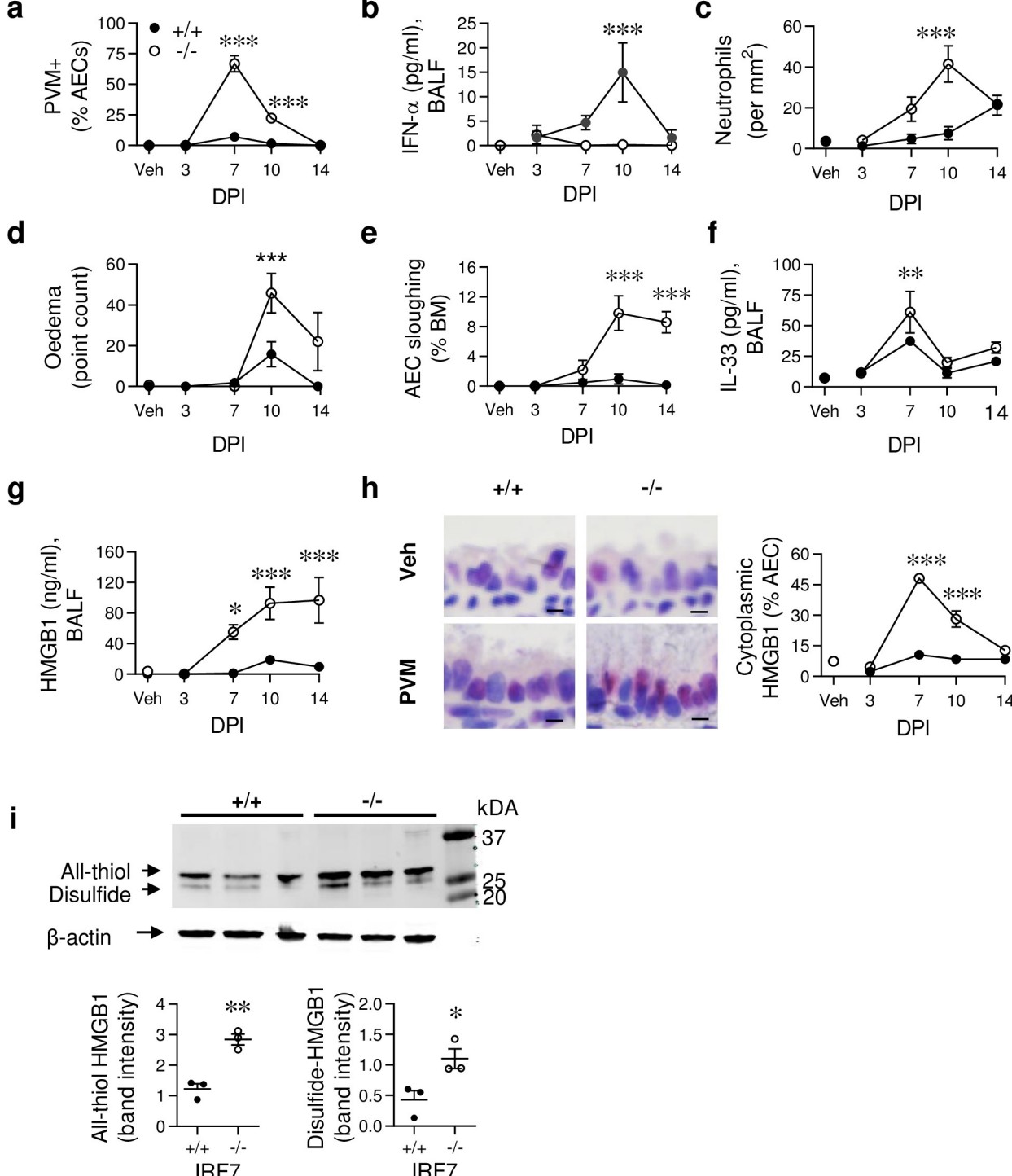

**Fig 1. IRF7 deficiency impairs antiviral immunity, heightening tissue damage and alarmin release.** WT (IRF7[+/+], closed circle) and IRF7[-/-] mice (open circle) were inoculated with PVM or vehicle at postnatal day 7 and samples collected at 3, 7, 10 and 14 days post infection (dpi). (a) Quantification of PVM+ airway epithelial cells (AECs). (b) IFN-α protein expression in BALF. (c) Ly6G+ neutrophils in lung sections. (d) Oedema. (e) AEC sloughing as a proportion of basement membrane (BM) length. (f) IL-33 and (g) HMGB1 protein expression in BALF. (h) Representative micrograph (x1000 magnification) of HMGB1immunoreactivity at 7 dpi (left panel) and quantification of cytoplasmic HMGB1 in AECs (right panel). Bars, 5 μm. (i) Electrophoretic mobility of lung homogenate loaded onto a 12% SDS-PA gel, and revealed by western blotting using polyclonal antibody against HMGB1 in PVM-infected WT (+/+) and IRF7[-/-] (-/-) mice at 10 dpi (top). Quantification of band intensity (bottom). Data are representative of $n$ = 2 experiments with 4–8 mice in each group and are presented as mean ± SEM (a-g) or scatter plot (i). Data

were analysed by Two-way ANOVA with Tukey post hoc test (a-g) or T-test (i); *, P < 0.05; **, P < 0.01; ***, P < 0.001 compared with the WT control group.

dpi, lung but not airway ILC2s were significantly higher in IRF7$^{-/-}$ mice compared to WT mice (S1E Fig). Other immune cells such CD4+ and CD8+ T cells also expressed dsRed, however their numbers were negligible as compared to ILC2s in infected IRF7-/- mice (Fig 2D). In both

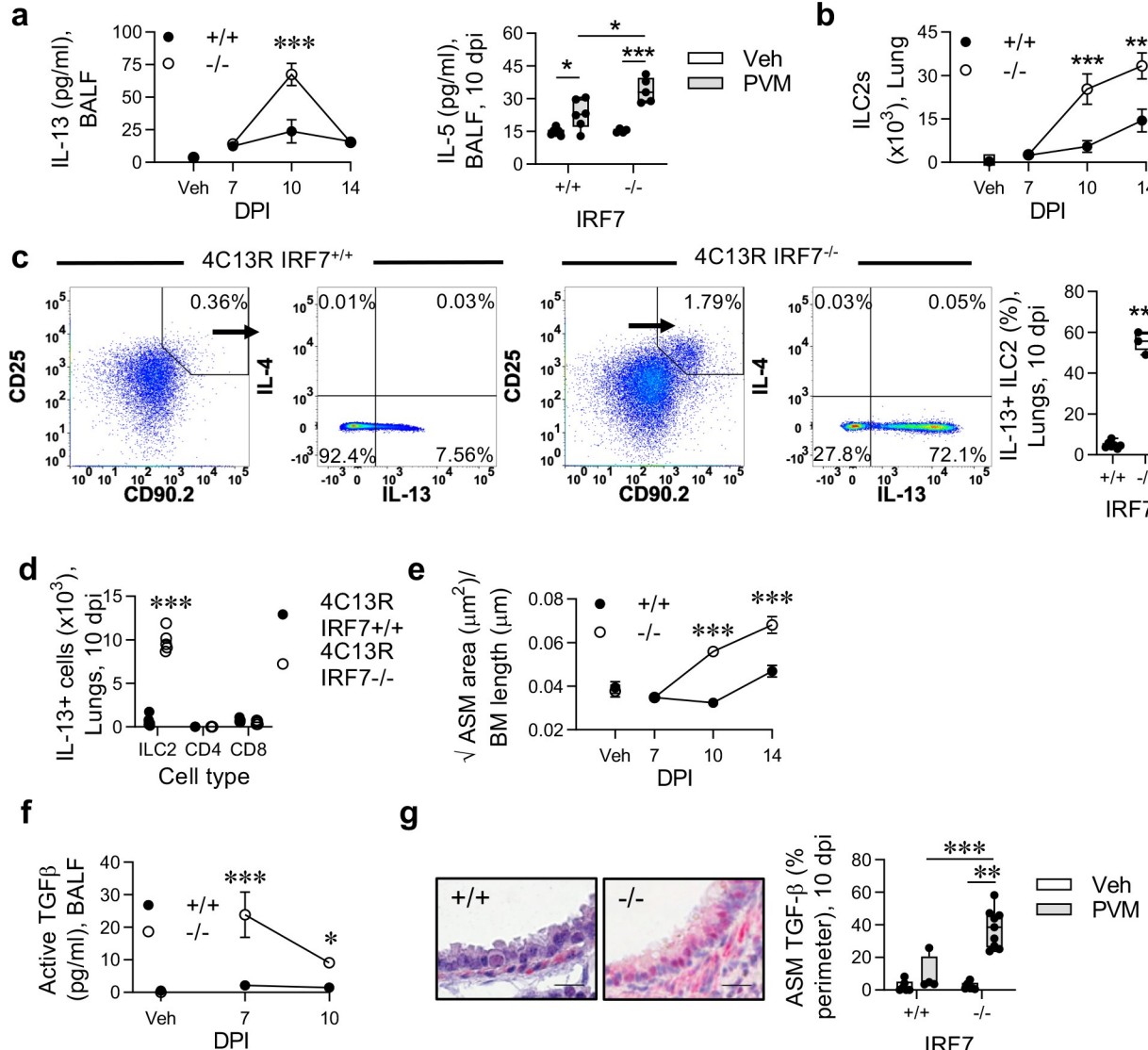

**Fig 2. IRF7 deficiency predisposes to Pneumovirus-induced type-2 inflammation and ASM growth in early-life.** WT (IRF7$^{+/+}$) and IRF7$^{-/-}$ mice were inoculated with PVM at postnatal day 7 and endpoints assessed at the times indicated. (a) IL-13 and IL-5 protein (10 dpi) expression in BALF. (b) ILC2 numbers in the lung (c) Expression of IL-13 (dsRed) and IL-4 (AmCyan) by ILC2 in the lungs of 4C13R IRF7$^{+/+}$ and 4C13R IRF7$^{-/-}$ mice at 10 dpi. Numbers in quadrants show the percentages of gated cells. Percentage of IL-13-expressing ILC2 (right panel). (d) Numbers of IL-13-producing ILC2, CD4$^{+}$ and CD8$^{+}$ cells in left lung lobe at 10 dpi. (e) ASM area. (f) Active TGFβ protein expression in BALF. (g) Representative micrograph (x400 magnification) of TGF-β1 immunoreactivity (red) in the lung at 10 dpi. Bars, 20 μm (left panels). Quantification of TGF-β1 expression by ASM cells (right panel). Data are representative of *n* = 2 experiments with 4 to 8 mice in each group and are presented as mean ± SEM (a [left], b, e) or as box-and-whisker plots showing quartiles (boxes) and range (whiskers; a [right], c [right], d, f). Data were analysed by Two-way ANOVA with Tukey post hoc test (a [left], b, e) or One-way ANOVA with Dunnett post hoc test (a [right], f [right]) or T-test (c [right], d); *, P < 0.05; **, P < 0.01; ***, P < 0.001 compared with the WT control group or as indicated.

strains, ILC2s were largely negative for AmCyan (reporting for IL-4; Fig 2C). Changes to the size of ASM area can occur in the airway wall of wheezy, pre-asthmatic infants [18]. We found an increase in ASM area and proliferation in IRF7$^{-/-}$ compared to WT infected mice (Fig 2E and S1F Fig), suggesting that ASM alterations are initiated in response to a severe respiratory infection in early-life. TGF-β is a potent ASM mitogen [32]. We found that active TGF-β levels in BALF were significantly higher in IRF7$^{-/-}$ mice compared to WT mice at both 7 and 10 dpi (Fig 2F). Probing the lung tissue for TGF-β expression by immunohistochemistry revealed the airway epithelium and ASM (clearly evident by morphology) to be a rich source of TGF-β following infection, and that its expression in ASM cells was significantly greater in IRF7$^{-/-}$ mice compared to WT mice (Fig 2G). Together, these data demonstrate that, in a susceptible host, a respiratory viral infection can lead to the induction of type-2 inflammation and ASM remodelling in early life.

## Secondary Pneumovirus infection or cockroach allergen exposure induces an asthma-like phenotype in IRF7-deficient mice

Frequent viral infections are associated with progression to asthma [33], which we have previously modelled in TLR7$^{-/-}$ mice and pDC-depleted mice [28,34]. Here, we identified that reinfection of IRF7$^{-/-}$ mice (at 7 weeks of age) induces an increase in ASM area (Fig 3A), and that this phenotype persists for at least 8 weeks (S2A Fig). Viral challenge also increased collagen deposition in the airway mucosa (S2B Fig) and increased airway hyperreactivity in the IRF7$^{-/-}$ mice (Fig 3B). Although this response was maximal at 7 days, it remained significantly elevated at 21 days post viral challenge (S2C Fig), presumably as a consequence of changes to the airway wall architecture. These responses were not associated with viral load, since the virus was not detectable by immunohistochemistry, even at 1 dpi. However, as with the primary infection, the induction of airway remodelling coincided with increased type-2 inflammation (IL-13 production, airway eosinophilia, mucus hypersecretion), which waned by 21 dpi (Fig 3C–3E). Again, the type-2 inflammation was preceded by an increase in IL-33 and HMGB1 expression in BALF of the IRF7$^{-/-}$ mice (Fig 3F). Additionally, we identified that IRF7$^{-/-}$ mice that had experienced severe bronchiolitis in infancy were predisposed to develop experimental allergic asthma in later life following exposure to low-dose cockroach allergen extract (CRE; S2D and S2E Fig). In contrast, PVM/CRE co-exposed WT mice did not develop asthma-like pathologies (S2E Fig). Importantly, in the absence of the viral infection in early-life, exposure to CRE alone did not induce experimental asthma in IRF7$^{-/-}$ mice.

## Neutralisation of HMGB1 or IL-33 diminishes type-2 inflammation and ASM growth in IRF7$^{-/-}$ mice

IL-33 has been implicated in ILC2 proliferation and airway remodelling; however, the contribution of HMGB1 to these processes remains an open question. To investigate this, we treated IRF7$^{-/-}$ mice with anti-HMGB1 (clone 2G7), which neutralises both all-thiol and disulphide HMGB1, or an isotype-matched control (S3A Fig), then killed the mice at 10 dpi when the pathologies associated with asthma onset were most prominent. For comparative purposes, we treated a separate group of mice with anti-IL-33. Anti-HMGB1 decreased HMGB1 but not IL-33 levels in BALF, whereas blockade of IL-33 lowered both HMGB1 and IL-33 (Fig 4A and 4B), suggesting that IL-33 operates upstream of HMGB1. Anti-HMGB1 and anti-IL-33 treatment of IRF7$^{-/-}$ mice decreased ILC2 numbers in the lung, type-2 cytokine responses in BALF, and airway eosinophilia (Fig 4C and 4D and S3B Fig). Moreover, both treatments arrested ASM growth (Fig 4E), an effect that was associated with lower TGF-β expression in the ASM layer (Fig 4F). Of note, treatment of IRF7$^{-/-}$ mice with glycyrrhizin, a putative HMGB1

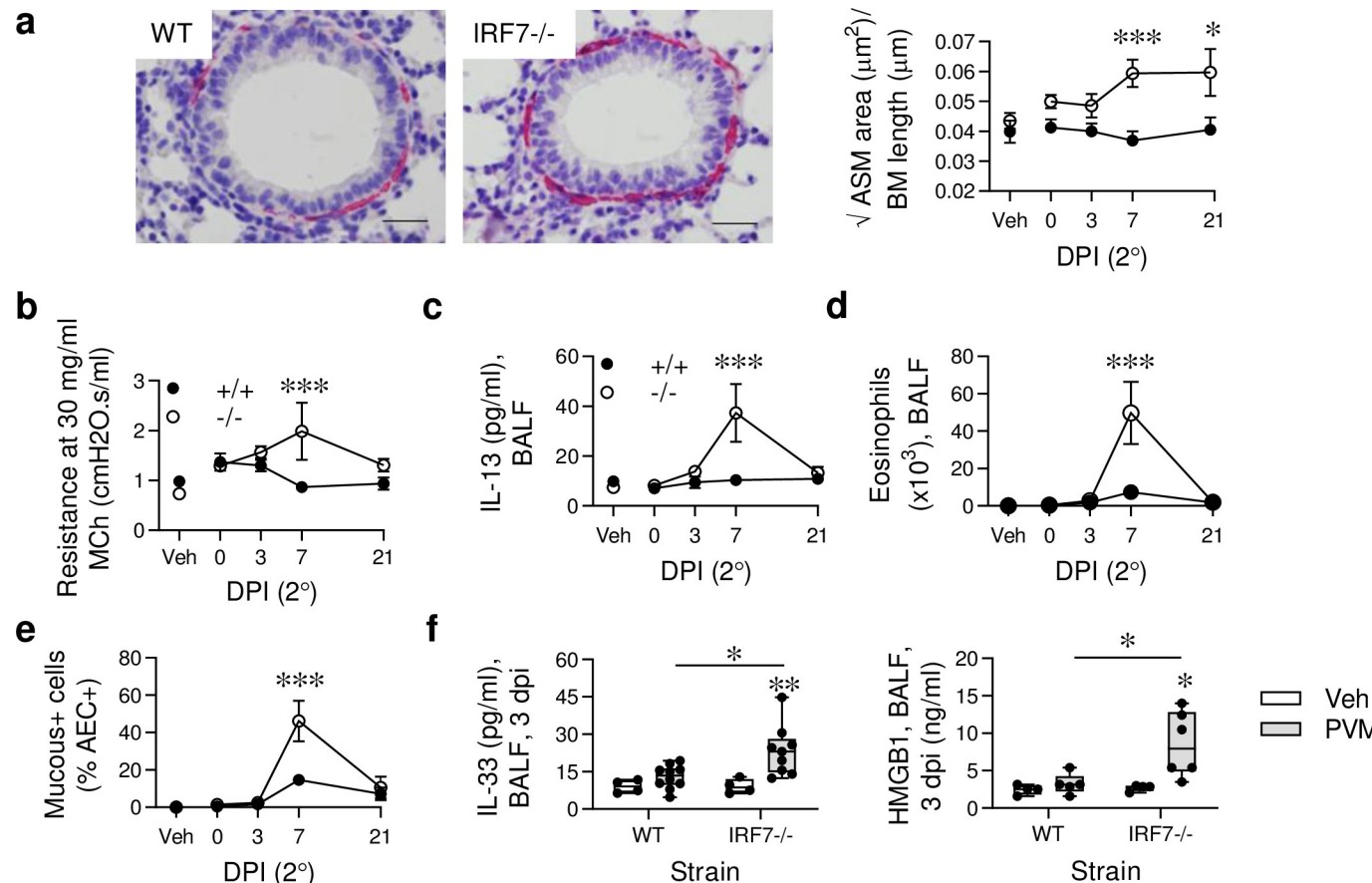

**Fig 3. Secondary Pneumovirus infection promotes an asthma-like pathology in IRF7-deficienct mice.** WT (IRF7$^{+/+}$) and IRF7$^{-/-}$ mice were inoculated with PVM at postnatal day 7 and re-infected 6 weeks later. Samples were collected prior to re-infection (t = 0) and at 3, 7 and 21 dpi. (a) Representative micrograph (x400 magnification) of α-smooth muscle actin immunoreactivity (red) in the lung at 7 dpi and ASM area quantification. Bars, 20 μm. (b) AHR at 30 mg/ml of methacholine. (c) IL-13 protein levels in BALF. (d) Total eosinophil numbers in BALF. (e) Mucus-producing airway epithelial cells (AECs). (f) IL-33 (left) and HMGB1 (right) protein expression in BALF at 3 dpi. Data are representative of $n$ = 2 experiments with 4–9 mice in each group and are presented as mean ± SEM (a ([right], b—e) or as box-and-whisker plots showing quartiles (boxes) and range (whiskers; f). Data were analyzed by Two-way ANOVA with Tukey post hoc test (a ([right], b—e) or One-way ANOVA with Dunnett post hoc test (f); *, P < 0.05; **, P < 0.01; ***, P < 0.001 compared with the WT control group.

antagonist, reproduced the findings observed with anti-HMGB1 (S3C–S3I Fig). Collectively, these data indicated that in a host with impaired with antiviral immunity (i.e. IRF7$^{-/-}$ mice), IL-33 and HMGB1 contribute to increased ILC2 numbers, type-2 cytokine production and the development of ASM remodelling in response to an early-life viral infection.

## Activated ILC2s from IRF7$^{-/-}$ mice induce ASM proliferation *in vitro* in an HMGB1-dependent manner

ILC2s contribute to tissue remodelling by promoting fibrosis, collagen deposition, and epithelial cell regeneration [12–15], however, they have yet to be implicated in the expansion of ASM area, a cardinal feature of asthma. To confirm a role for ILC2s, we next developed an *in vitro* co-culture assay to test whether ILC2s directly induce ASM cell proliferation. Primary tracheal ASM cells were isolated from neonatal WT mice, loaded with a proliferation dye and cultured in an FCS-low (0.25%) media prior to stimulation. Testing the proliferative effects of IL-33 and IL-13 (relative to 10% FCS) revealed that IL-33 is a poor mitogen (S4A Fig), whereas IL-13 is a potent mitogen; with an EC$_{50}$ below the level of IL-13 detected in BALF (Fig 2A). To

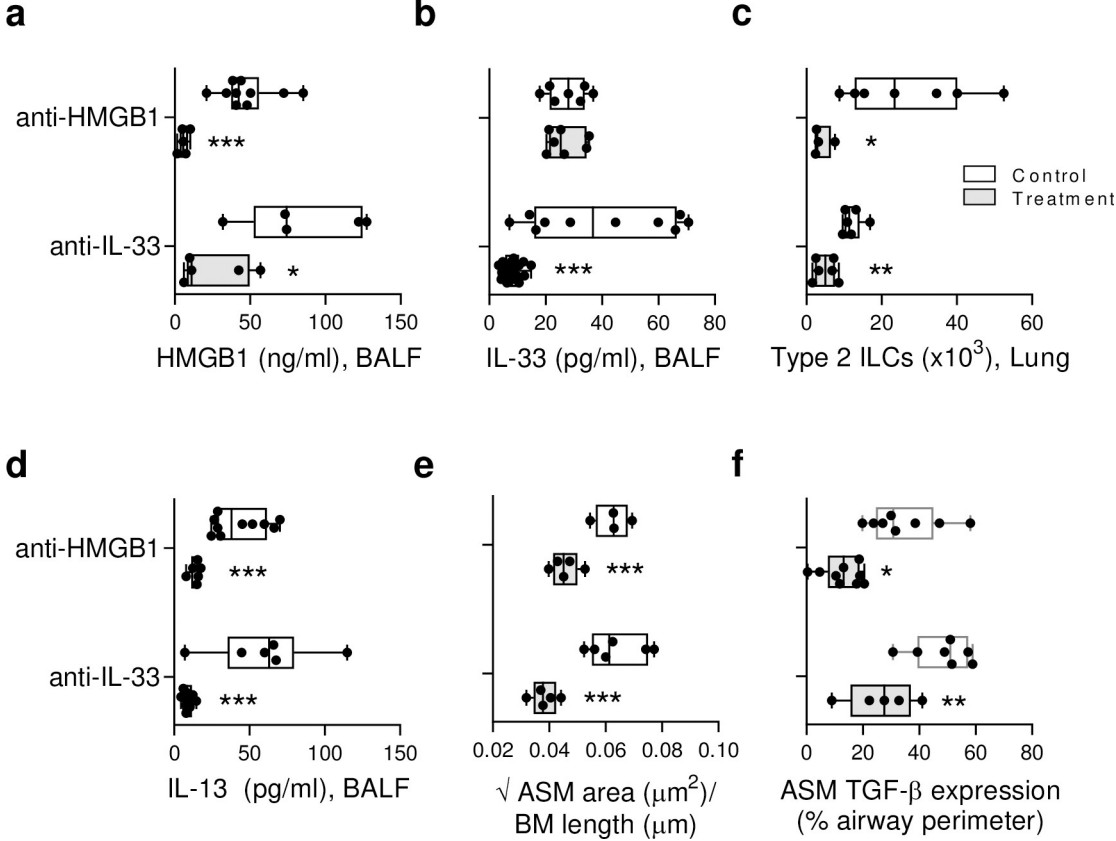

**Fig 4. Anti-HMGB1 diminishes type-2 inflammation and ASM growth in neonatal IRF7[-/-] mice.** IRF7[-/-] mice were inoculated with PVM as described in Fig 2, and treated with anti-HMGB1 or anti-IL-33 as described in S3 Fig. (a) HMGB1 and (b) IL-33 protein expression in BALF. (c) ILC2 numbers in lung. (d) IL-13 protein expression in BALF. (e) ASM area. (f) TGF-β1 expression by ASM cells. Data are representative of *n* = 2 experiments with 4–8 mice in each group and are presented as box-and-whisker plots showing quartiles (boxes) and range (whiskers). Data were analysed by t-test; *, P < 0.05; **, P < 0.01; ***, P < 0.001.

directly assess whether ILC2s promote ASM proliferation, we FACS-purified ILC2s from IRF7[-/-] mice lungs at 10 dpi (S4B Fig). As expected, these ILC2s produced vast amounts of IL-5, IL-9 and IL-13, but not IL-4, IFN-γ or IL-17A, when stimulated with IL-2/IL-33 but not IL-2 alone (S4C Fig). In contrast to a recent report suggesting that stimulated ILC2s from adult IRF7[-/-] mice produce less type-2 cytokine [35], we were unable to replicate this finding; IL-2/IL-33-stimulated WT and IRF7[-/-] ILC2s produced equal amounts of IL-5 and IL-13 (S4D Fig). When co-cultured at a 1:5 ratio with ASM cells, the IL-2/IL-33-stimulated ILC2s, but not IL-2/IL-33 or unstimulated ILC2s alone, were able to promote ASM cell proliferation (Fig 5A, upper panel). This response was associated with the production of IL-13 (Fig 5A), and was attenuated when IL-13 was neutralised with sol-IL-13Rα2 (Fig 5A). Active TGF-β was significantly elevated in the IL-2/IL-33-activated ILC2/ASM co-cultures and significantly decreased when IL-13 was neutralised (Fig 5A). As anti-HMGB1 treatment attenuated ASM growth and TGF-β expression *in vivo* (Fig 4), we questioned whether HMGB1 contributed to ILC2-mediated ASM proliferation in the co-culture system. Interestingly, HMGB1 levels were elevated in the IL-2/IL-33-stimulated ILC2-ASM co-cultures, and were diminished when IL-13 was neutralised (Fig 5A), implicating a role for HMGB1 downstream of IL-13. When HMGB1 was neutralised (anti-HMGB1) or antagonised from activating TLR4 (LPS-RS), ILC2-mediated ASM proliferation and the production of active TGF-β were diminished (Fig 5A).

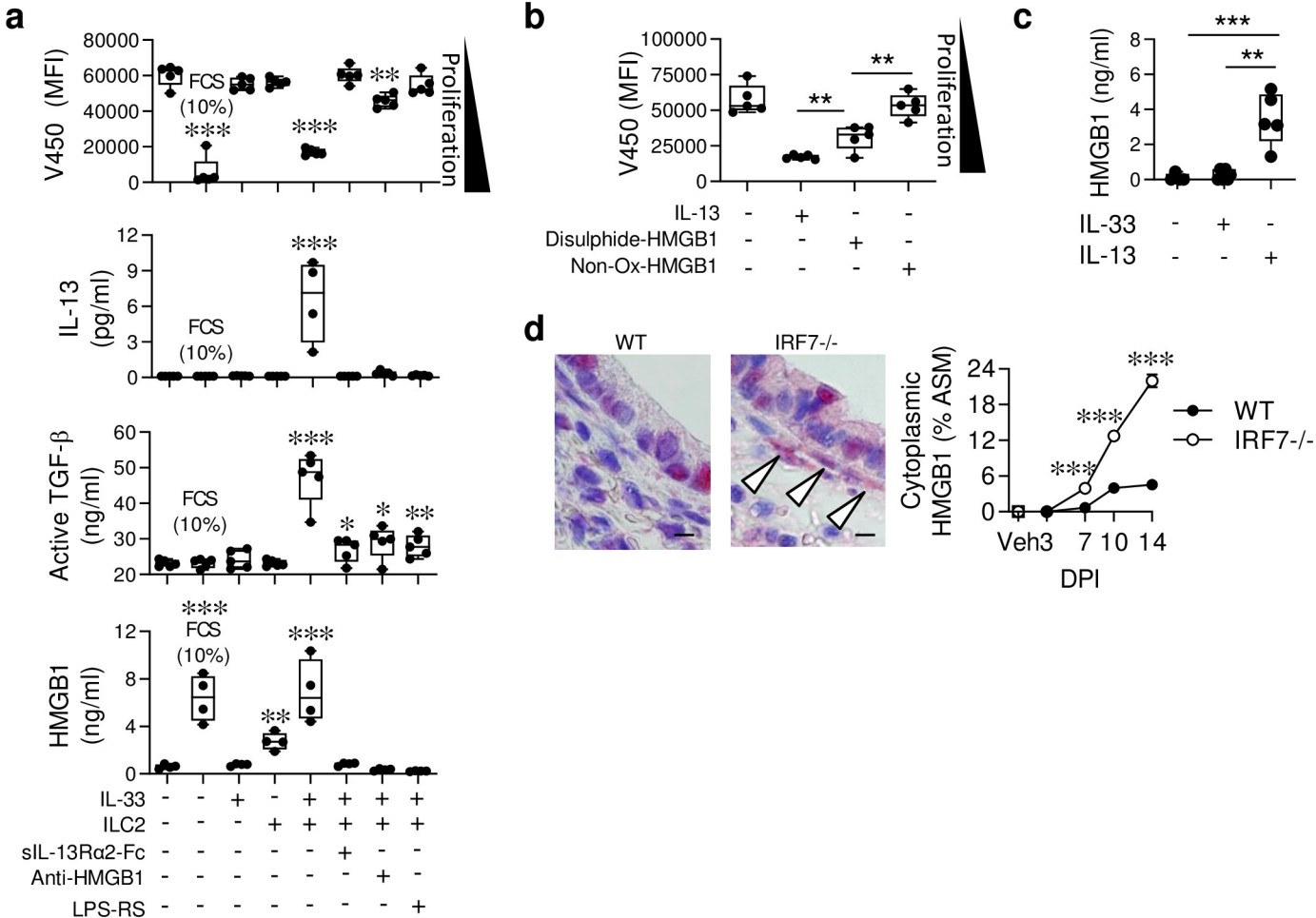

**Fig 5. Activated ILC2s induce ASM proliferation *in vitro* in an HMGB1-dependent manner.** (a-b) Primary ASM cells were isolated from naïve WT neonatal mice and cultured with ILC2s (purified from the lungs of IRF7[-/-] mice at 10 dpi) in the presence of various stimuli as indicated. (a) ASM proliferation (indicated by decreasing median fluorescence intensity (MFI)), IL-13, active TGFβ and HMGB1 concentration in the culture supernatant. * Compared to ASM cells cultured in the absence of ILC2s or cytokine stimulation. (b) ASM proliferation in response to IL-13 (3 ng/ml), disulphide HMGB1 (30 ng/ml), or a non-oxidisable mutant HMGB1 (30 ng/ml). * Compared to IL-13-stimulated ASM cells with disulphide-HMGB1-stimulated ASM cells or disulphide-HMGB1-stimulated ASM cells with non-oxidisable mutant HMGB1-stimulated ASM cells as indicated. (c) HMGB1 protein expression in the supernatant of ASM cells cultured in the presence of IL-33 or IL-13 as indicated. (d) Representative micrograph (x1000 magnification) of HMGB1 immunoreactivity (red) in the lung at 10 dpi, arrows indicate expression by ASM cells. Bar, 5 μm. Quantification of HMGB1 expression by ASM cells. For panel a-d, data are representative of one experiment, performed twice with similar results. For panel e, data are representative of *n* = 2 experiments with four to eight neonates in each group. Data are presented as box-and-whisker plots showing quartiles (boxes) and range (whiskers; a-d) or as mean ± SEM (e [right]). Data were analysed by One-way ANOVA with Dunnett post hoc test (a-d) or Two-way ANOVA with Tukey post hoc test (e ([right]; *, P < 0.05; **, P < 0.01; ***, P < 0.001 compared to the respective control group.

Unexpectedly, both of these treatments also attenuated the production of IL-13, indicating the existence of a feed-forward amplification loop between HMGB1/TLR4 and IL-13 (Fig 5A).

The requirement for TLR4 inferred a role for disulphide-HMGB1 [30,36]. Consistent with this theory, stimulation of ASM cells with disulphide HMGB1, but not the non-oxidisable mutant all-thiol HMGB1 [30], induced their proliferation (Fig 5B). As IL-13 but not IL-33 stimulated ASM cells produced HMGB1 (Fig 5C), and because IL-13-induced ASM cell proliferation was attenuated by TLR4 or RAGE antagonism (S4E Fig), this implicated a key role for ASM cell-derived HMGB1. In light of these finding, we re-examined the lung sections probed with anti-HMGB1. Whereas vehicle-inoculated mice did not express HMGB1 in the ASM, its expression increased incrementally over the course of infection in the ASM bundles of IRF7[-/-]

but not WT mice (Fig 5D), similar to the pattern of ASM growth (Fig 2E). In summary, our findings suggest that in a genetically susceptible host (i.e. IRF7[-/-] mice in this context), ILC2-produced IL-13 induces the release of HMGB1, leading to the activation of TLR4 and/or RAGE signalling, which in turn, increases TGF-β production, ASM proliferation, and amplifies the production of IL-13.

### ILC2-derived HMGB1 in IRF7[-/-] mice acts in an autocrine manner to amplify IL-13 production

Demonstrating that ASM cells release HMGB1 did not rule out the possibility that ILC2-derived HMGB1 might contribute to ASM proliferation. Following IL-2/IL-33 stimulation of ILC2s purified from IRF7[-/-] mice, HMGB1 was elevated in the cell cytoplasm and detected in the supernatant (Fig 6A [left]), and this release was significantly diminished upon IL-13 neutralisation (Fig 6A [right]). Together with our findings from the ILC2-ASM cell co-culture system (Fig 5B), where anti-HMGB1 ablated IL-13 levels, these data suggested a feedforward loop whereby IL-13-induced HMGB1 amplifies the production of IL-13. Consistent with this theory, IL-2/IL-33-stimulated ILC2s produced IL-13 in a biphasic manner, with the second phase occurring after the peak of HMGB1 release (Fig 6B). As we had previously shown that RAGE[-/-] mice generate attenuated type-2 inflammatory responses to both viral- or allergen-triggered experimental asthma [6,37], we speculated that HMGB1/RAGE ligation promotes type-2 cytokine production by ILC2s. To determine whether ILC2s express RAGE, we used RAGE[-/-] mice in which the functional *Ager* gene has been replaced by *GFP*. Compared to non-stained WT ILC2s (grey), ILC2s derived from RAGE[-/-] mice (black) were GFP bright, indicating that ILC2s express RAGE at steady state (Fig 6C). To assess the role of HMGB1 directly, we stimulated lung ILC2s in culture with disulphide-HMGB1 or non-ox-HMGB1 in the presence of IL-2 for 6 days. Non-Ox-HMGB1 but not disulphide-HMGB1 induced an increase in IL-5 and IL-9 production. IL-13 production was numerically higher but this effect was not significant (Fig 6D). To determine whether RAGE or HMGB1 contribute to IL-33-induced type-2 cytokine expression by ILC2s, we stimulated ILC2s (isolated from the lungs of infected 4C13R IRF7[-/-] mice) with IL-2/IL-33 for 4 days in the presence of anti-HMGB1, LPS-RS (a TLR4 antagonist) or FPS-ZM1 (a RAGE antagonist) [38]. The increased intensity of the dsRed signal (IL-13 expression) following IL-2/IL-33 stimulation was diminished by anti-HMGB1 or RAGE antagonism, but not TLR4 antagonism (Fig 6E), implicating an ILC2 intrinsic HMGB1/RAGE signalling pathway that amplifies IL-13 production. Alterations in ILC2 cell death or proliferation amongst the treatment groups did not account for this effect (S4F Fig). In addition to attenuating IL-13 expression, anti-HMGB1 and RAGE antagonism ablated the production of IL-5 and IL-9 (Fig 6F) in the culture supernatant, suggesting that HMGB1-mediated activation of RAGE contributes to the regulation of ILC2 type-2 cytokine production.

Finally, to confirm that RAGE contributes to the development of type-2 inflammation in PVM-infected IRF7[-/-] mice, we administered the RAGE antagonist, FPS-ZM1. Consistent with the lack of effect of RAGE deficiency on ILC2 proliferation (S4F Fig), lung ILC2 numbers were unaffected. However, RAGE antagonism significantly decreased IL-5 levels in BALF, diminished airway eosinophilia, and critically, prevented aberrant ASM growth in infancy (Fig 6G–6J).

## Discussion

Severe viral bronchiolitis is an independent risk factor for asthma. To elucidate novel pathogenic processes that link the two diseases, we simulated a clinically relevant gene-environment interaction by infecting IRF7[-/-] mice with PVM. In the absence of IRF7, viral burden increased

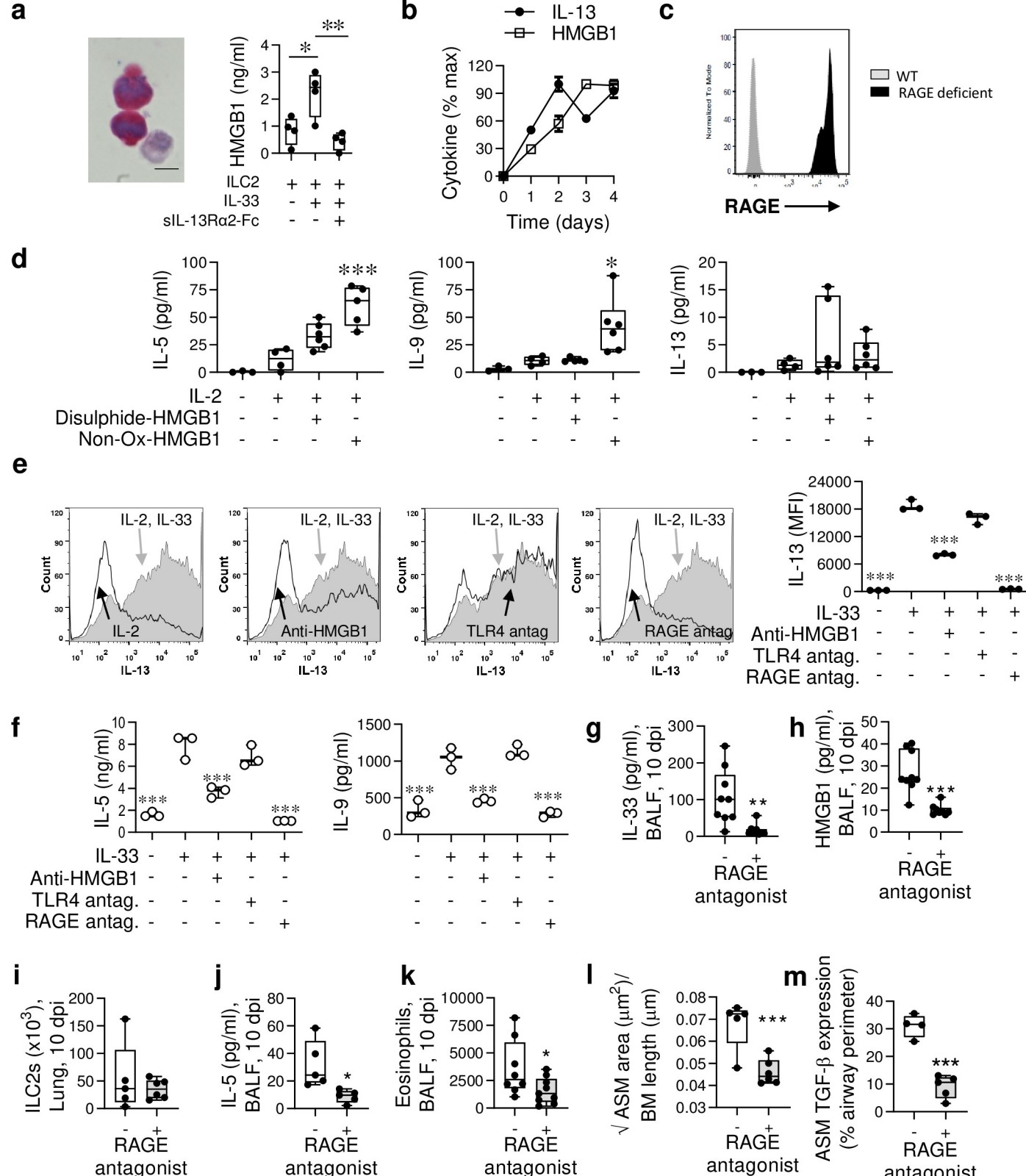

**Fig 6. HMGB1/RAGE axis amplifies ILC2 type-2 cytokine production and contributes to ASM growth *in vivo*.** (a [left]) Lung ILC2s were purified from IRF7[-/-] mice at 10 dpi and cultured with IL-2 for 4 days. Representative micrograph (x1000 magnification) of HMGB1 expression (red) in IL-33 stimulated ILC2s. Bar,

5 μm. (a [right]) HMGB1 levels in the culture supernatant after a 4 day culture +/- IL-33 (3 ng/ml) and +/- sIL-13Rα2 (5 μg/ml) as indicated. (b) IL-13 and HMGB1 protein expression by ILC2s. (c) RAGE expression detected by flow cytometry in lung ILC2s. (d) IL-5, IL-9 and IL-13 protein expression by lung ILC2s purified from IRF7$^{-/-}$ mice cultured in IL-2 +/- difulphide-HMGB1 (30 ng/ml) or non-oxidisable mutant HMGB1 (30 ng/ml) as indicated. * Compared to IL-2-stimulated ILC2s. (e) Lung ILC2s purified from 4C13R IRF7$^{-/-}$ mice were cultured in IL-2 +/- IL-33, and treated with anti-HMGB1 (Clone 2G7, 5 μg/ml), TLR4 antagonist (LPS-RS 1 μg/ml) and RAGE antagonist (FPS-ZMI; 300 nM) as indicated. DsRed (IL-13) intensity was quantified by flow cytometry. * Compared to IL-2/IL-33-stimulated ILC2s. (f) IL-5 and IL-9 production by ILC2s, treated as described in panel e. (g-m) IRF7$^{-/-}$ mice were inoculated with PVM as described in Fig 2, and treated daily from 3–10 dpi with a RAGE antagonist (FPS-ZM1). (g) IL-33 and (h) HMGB1 protein expression in BALF. (i) Lung ILC2s, (j) IL-5 protein in BALF, (k) Eosinophils in BALF, (l) ASM area, and (m) Quantification of TGF-β1 expression by ASM cells. For panel a-f, data are representative of one experiment, performed twice with similar results. For panel g-m, data are representative of $n$ = 2 experiments with 4–9 mice in each group and are presented as box-and-whisker plots showing quartiles (boxes) and range (whiskers; a [right], d, g-m) or as mean ± SEM (b) or as scatter plot (e [right]-f). Data were analysed by One-way ANOVA with Dunnett post hoc test (a, d, e [right]-f) or T-test (g-m); *, P < 0.05; **, P < 0.01; ***, P < 0.001 compared to the respective control group.

substantially in the airway epithelium, leading to epithelial sloughing (i.e. cell death), HMGB1 release, expansion of IL-13-producing ILC2s, and ASM growth. Anti-HMGB1 decreased ILC2 numbers and IL-13 production, and attenuated ASM growth. Similarly, anti-HMGB1 attenuated ILC2-induced ASM proliferation in a co-culture model. In addition to the airway epithelium and underlying smooth muscle, we identified that ILC2 also release HMGB1 and that this contributes to an IL-13-induced auto-amplification loop via the activation of RAGE. Thus, our data suggest that by dampening type-2 inflammation and attenuating virus-associated ASM remodelling, biologics or small molecules targeting HMGB1 and/or RAGE will serve as novel preventatives for the treatment of asthma.

Several lines of evidence support the concept that impaired antiviral immunity in early-life is associated with the later development of asthma. To identify novel pathogenic mechanisms that occur consequent to defective antiviral host defence, we inoculated IRF7$^{-/-}$ neonatal mice and followed host-pathogen interactions in a temporal manner. In IRF7$^{-/-}$ mice, the peak of infection (7 dpi) was associated with increased nuclear-to-cytoplasmic HMGB1 translocation in the epithelium, increased IL-33 levels in the airway lumen, and the infiltration of neutrophils, which can process IL-33 to a more biologically active form [39]. This response was followed (10 dpi) by increased epithelial sloughing, elevated HMGB1 in the airway, ILC2 expansion, and growth of the underlying ASM layer. Of the lung ILC2s, approximately half were producing IL-13, presumably as a consequence of the cytokine milieu, i.e. high alarmin (IL-33/HMGB1) concentration in the absence of counter regulatory signals from type I IFNs or IL-12 [34,40–42]. HMGB1 neutralisation attenuated the type-2 inflammatory response and arrested ASM growth, suggesting a role for HMGB1 and ILC2 in the development of ASM remodelling.

To implicate ILC2s directly, we developed an ILC2-ASM cell co-culture model. Using this system, we demonstrated that ILC2-derived IL-13 induces ASM cell proliferation, and that this effect is associated with increased ASM cell TGF-β expression (also elevated *in vivo*). Although IL-13 is known to mediate its fibrotic effects via TGF-β[43], our findings suggest that, in ASM cells at least, HMGB1 appears to act as an intermediary in this process. Of note, ASM expression of HMGB1 is elevated in lung biopsies of patients with asthma [44]. Consistent with a functional role for HMGB1, TLR4 blockade lowered active TGF-β expression, attenuating ILC2-induced ASM proliferation. This suggested a linear pathway whereby IL-13 produced by ILC2s induces ASM cells to release HMGB1, which in turn, activates TLR4 to induce TGF-β driven proliferation. However, neutralisation of HMGB1 in the co-culture system affected the production of 'upstream' IL-13, raising three non-mutually exclusive possibilities: (i) ASM-derived HMGB1 activates ILC2s to support IL-13 production, (ii) ASM-derived HMGB1 activates TLR4 on ASM cells, which in turn, releases a secondary mediator that supports IL-13 production by ILC2, and (iii), HMGB1 is released by ILC2s and acts in an autocrine manner to amplify IL-13. In support of the latter possibility, we demonstrated that IL-2/

IL-33-stimulated ILC2s (in the absence of ASM cells) produce HMGB1 downstream of IL-13, and that the neutralisation of HMGB1 or blockade of RAGE, but not TLR4, attenuates the production of IL-13 by ILC2 (as assessed by dsRed intensity and IL-13 protein production). Similarly, antagonism of the HMGB1/RAGE axis ablated IL-5 and IL-9 production, thus identifying a novel autocrine role for HMGB1 in the production of type-2 cytokines by ILC2s. We also demonstrated that non-oxidisable HMGB1 induces type-2 cytokine production by ILC2s. A recent paper also implicated a role for RAGE downstream of IL-13 (and IL-4, IL-5), although the nature of the RAGE ligand remained unclear, with HMGB1 not tested exhaustively [45]. Consistent with a role for HMGB1/RAGE in mediating type-2 inflammation, we found that RAGE antagonism dampened type-2 inflammation and ASM remodelling in infected IRF7[-/-] mice. However, RAGE antagonism was not as effective as anti-HMGB1, suggesting that HMGB1 exerts RAGE-independent pathways. Indeed, it was also noteworthy that TLR4 antagonism diminished IL-13 production in the ILC2-ASM co-cultures, but had no effect on IL-13 production when the ILC2s were cultured alone, suggesting that the activation of TLR4 on ASM cells induces a factor which feeds back to the ILC2s. One possibility is TGF–β, which supports lung ILC2 [46]. If such bidirectional crosstalk between inflammatory cells and structural cells sustains the inflammatory response and underlies disease chronicity, then breaking these circuits offers an opportunity for new and effective therapeutic interventions.

Asthma is a polygenic disorder, underpinned by gene-environment interactions. As individual SNP effects on disease risk are small, we elected to use knockout mice, which we have used previously to identify novel pathogenic mechanisms [34,37,47]. We believe this approach is preferable to using an extremely high inoculum in WT mice that can induce non-physiological responses and increase the potential for false positives. However, by using IRF7[-/-] mice, we do not presume that our findings are relevant to all cases of viral bronchiolitis in infancy; indeed we agree with view that there are different endotypes of bronchiolitis [48], as is now accepted to be the case for asthma. On the other hand, we do not take the view that our findings are only amenable to individuals with a genetic variant affecting IRF7, not least since exacerbating children can be subdivided into IRF7[hi] and IRF7[lo] phenotypes based on gene expression [27]. In the *in vitro* ILC2: ASM cell co-culture model, we used exogenous IL-33 rather than encapsulate the lung cytokine microenvironment, and thus, there is no merit in exploring whether IRF7-sufficient ILC2s would promote ASM proliferation (i.e. the phenotype is IL-13 dependent, so an IL-33-stimulated IRF7-sufficient ILC2s will self-evidently induce ASM proliferation).

Taken together with our previous findings, and reports from other groups, it is increasingly apparent that HMGB1 is an important mediator in the pathogenesis of viral bronchiolitis and asthma [4–6,8,9,44]. Intriguingly, PVM infection of RAGE[-/-] mice induces viral bronchiolitis characterised by impaired antiviral immunity and ASM remodelling but not type-2 inflammation [37]. Notably, HMGB1 neutralisation or the backcrossing of RAGE[-/-] mice with TLR4[-/-] mice protects against the development of ASM remodelling, implicating disulphide-HMGB1/TLR4 activation in ASM remodelling. Our findings here support a role for the same ligand-receptor pairing in mediating ILC2-induced ASM growth. In contrast, HMGB1/RAGE activation appears more critical to the amplification of type-2 inflammation.

Collectively, our findings provide a mechanistic basis to explain the effectiveness of anti-HMGB1 treatment or RAGE deficiency in acute and chronic models of allergen- and viral-induced experimental asthma. HMGB1/RAGE contributed to ILC2-mediated type-2 cytokine production and type-2 inflammation, and ILC2-produced IL-13 activates a HMGB1/TLR4/RAGE signalling pathway in ASM cells leading to increased TGF-β expression and ASM remodelling.

## Methods

### Experimental models

**Animals.** IRF7[-/-] (provided by Dr. Tadatsugu Taniguchi, University of Tokyo), RAGE[-/-] (provided by Dr. Ann Marie Schmidt, NYU Langone Medical Centre) and 4C13R mice (provided by Dr. William Paul, NIH) [49] were imported and rederived at The University of Queensland. Mice were housed under specific pathogen–free conditions, on a 12 hour controlled day/night cycle with food and water available *ad libitum* in the University of Queensland or QIMR Berghofer Medical Research Institute, Australia. Mice were time-mated and litter sizes (4–8 mixed genders pups) standardized at postnatal day 3. Neonatal mice were genotyped using standard PCR and the following primers: IRF7[-/-] mice: mutant 5'-TCGTGCTT TACGGTATCGCCGCTCCCGATTC-3', common 5'-AGTAGATCCAAGCTCCCGGCTA AGTTCGTAC-3', WT 5'-GTGGTACCCAGTCCTGCCCTCTTTATAATCT-3'; 4C13R mice: forward 5′-GCTCCAAGGTGTACGTGAAG-3′ and reverse 5′-GCTTGGAGTCCACGTAG TAG-3′; and RAGE[-/-] mice: forward 5'-CCTGGGTGCTGGTTCTTG-3' and revers 5'-CTGA GGTCCGTGGCTAGG-3'. Stocks of pneumonia virus of mice (PVM) strain J3666 were prepared as described previously [50]. At postnatal day 7, mice were inoculated (intranasal route) with 2 pfu of PVM or vehicle (DMEM containing 10% v/v FCS) under isofluorane-induced anaesthesia. For viral challenge, mice were inoculated with 100 PFU PVM at 49 days of age. For cockroach extract (CRE) exposure at later life, mice were exposed to 1μg CRE at 49, 56, 63 and 70 days of age and euthanise 3 days later. In some experiments, mice were treated (i.p. route) with anti-IL-33 (30 μg/g body weight, Pfizer, Inc) or anti-HMGB1 (8 μg/g body weight, clone 2G7; provided by Dr. Kevin Tracey, The Feinstein Institute for Medical Research), isotype-matched control antibody (30 μg/g body weight), glycyrrhizin (50 μg/g body weight; Calbiochem) or RAGE antagonist (1 μg/g body weight, daily from 3 to 10 dpi; FPS-ZM1, Calbiochem).

### Ethics statement

The ARRIVE (Animal Research: Reporting of In Vivo Experiments) guidelines in the EQUA-TOR (Enhancing the Quality and Transparency of Health Research) Network library were followed for this report. All animal studies were performed in accordance with the Animal Care and Ethics Committees of the University of Queensland Animal Ethics Committee (AEC209/ 13) and QIMR Berghofer Medical Research Institute Animal Ethics Committee (P2319 A1706-609M) and are consistent with guidelines provided by the Australian National Health and Medical Research Council.

### Sample extraction and processing

Following euthanasia, BAL was performed and lung lobes excised and processed as follows: the left lung lobe was digested immediately for flow cytometry, the superior right lobe (for histological analysis) was fixed in formalin neutral buffer overnight before storage in 70% ethanol, the post-caval and inferior lobes (for protein analysis by ELISA) were pooled and snap-frozen, and the inferior right lobe (for quantification of mRNA expression) was snap frozen. All snap frozen lungs were stored at -80 ˚C.

### Flow cytometry

Single cell suspensions of lung cells were generated as described [28]. Following Fc block, cells were incubated with various antibodies to allow for immunophenotyping. These included: anti-mouse CD2-FITC (RM2-5; BD Biosciences Cat# 553111, RRID:AB_394632), CD4–AF488 (RM4-5; BD Biosciences Cat# 557667, RRID:AB_396779), CD11c–FITC (HL3; BD

Biosciences Cat# 553801, RRID:AB_395060), CD11b–AF488 (M1/70; BD Biosciences Cat# 557672, RRID:AB_396784), B220-AF488 (RA3-6B2; BD Biosciences Cat# 557669, RRID: AB_396781), CD3–AF488 (145-2C11; BD Biosciences Cat# 557666, RRID:AB_396778), CD19-AF488 (6D5; BioLegend Cat# 115524, RRID:AB_493339), Gr-1–AF488 (RB6-8C5; Bio-Legend Cat# 108419, RRID:AB_493480), NK1.1-AF488 (PK136; BioLegend Cat# 108717, RRID:AB_493184), CD45-BV421 (30-F11; BioLegend Cat# 103133, RRID:AB_10899570), ST-2-APC (DIH9; BioLegend Cat# 145305, RRID:AB_2561916), CD90.2-APC Cy7 (30-H12; Bio-Legend Cat# 105311, RRID:AB_313182), Sca-1-PE (D7; BioLegend Cat# 108107, RRID: AB_313344), CD25-BV650 (PC61; BioLegend Cat# 102037, RRID:AB_11125760), CD8-PE (53–6.7; BioLegend Cat# 100707, RRID:AB_312746), CD4-AF750 (RM4-5; Thermo Fisher Scientific Cat# MCD0427, RRID:AB_10371891). Cells were incubated with 7-AAD (eBioscience) for 5 min at 4˚C to exclude dead cells. To detect RAGE expression in cells, RAGE$^{-/-}$ mice with an EGFP insert was used, Cells were acquired using a LSR Fortessa X-20 (BD Biosciences), and live cells analysed using FACSDiva v8 (BD Biosciences) and FlowJo v8.8 (Treestar).

## Histology and Immunohistochemistry

Five micron thick lung tissue sections were stained with Periodic acid-Schiff or picrosirius red to enumerate mucous-secreting cells and collagen respectively. Immunohistochemistry for PVM G protein (gift from U. Buchholz, National Institutes of Health, Bethesda, MD), HMGB1 (polyclonal; Abcam Cat#ab18256, RRID:AB_444360), and Ly6G$^+$ neutrophils (1A8; BD Biosciences Cat# 551459, RRID:AB_394206) and quantification of immunoreactivity was performed as described previously [6,28,34,51]. ASM was detected using α-smooth muscle actin (1A4; Sigma-Aldrich Cat# A5691, RRID:AB_476746) monoclonal antibody and ASM area around the small airways (defined as a circumference <500 μm for neonates and <800 μm for mice >7 weeks old) was measured using Scanscope XT software and expressed as area per micrometer of basement membrane. AECs sloughing was measured using Scanscope XT software and expressed as percentage of epithelial denudation of airway basement membrane [34]. Oedema was assessed by point counting of fluid-filled airspaces [42]. To detect TGF-β1, tissue sections were pre-treated with 10% normal goat serum then incubated with anti-TGF-β1 (polyclonal; Abcam Cat# ab92486, RRID:AB_10562492) overnight, followed by incubation with anti-rabbit IgG-alkaline phosphatase (Sigma-Aldrich) for 60 min. Immunoreactivity was developed with Fast Red (Sigma-Aldrich), followed by counterstaining with Mayer's hematoxylin. TGF-β1 expression in the ASM is measured by the fraction of the airway perimeter that has an adjacent TGF-b immunoreactive ASM cell. For each mouse, 5 airways were measured, and the mean value used in the graph. To identify proliferating ASM cells, lung sections were sequentially incubated with mouse anti-α-SM actin ((1A4; Sigma-Aldrich Cat# A5691, RRID:AB_476746), goat anti-mouse AF488 (Thermofisher), anti-PCNA-biotin (PC10; Thermo Fisher Scientific Cat# 13–3940, RRID:AB_2533017), and streptavidin-AF647 (Thermofisher), then counter-stained with DAPI (Sigma-Aldrich). To enumerate airway eosinophils, a cytospin of BAL cells was generated then incubated with May-Grünwald Giemsa stain. To visualise airway epithelial basement membrane thickening, sections were incubated with picrosirius red, and the stained area quantified using ImageJ (NIH). Mucus was stained with periodic acid–schiff staining and scored as the percentage of mucus-secreting AECs. Researchers who performed histological analysis and scoring were blinded to the sample identity.

## Measurement of protein expression

Cytokine concentration was measured by ELISA according to the manufacturer's protocol. The sensitivity of the assay (or where unavailable, the lowest standard used) is provided in

parenthesis. IFN-α (2.38 pg/ml; PBL assay), IL-33 (15.6 pg/ml), IL-13 (1.5 pg/ml; R&D Systems), IL-9 (1.8 pg/ml), IL-12p40 (7.8 pg/ml, IL-17A (8 pg/ml), active-TGF-β1 (2.3 pg/ml) and IFN-γ (15 pg/ml; Biolegend), IL-4 (2 pg/ml), and IL-5 (4 pg/ml; BD Biosciences) and HMGB1 (0.78 ng/ml; Chondrex).

## Western Blot

Total protein in lung homogenates was determined using the Pierce BCA protein assay kit (Thermo Fisher Scientific). Equivalent amounts of protein were loaded and separated on 4–20% Mini-PROTEAN TGX Precast Protein gels (Bio-Rad Laboratories) followed by transfer on to a PVDF membrane (Bio-Rad Laboratories). After blocking with 5% skim milk in 0.1% Tween 20/TBS, the membranes were probed with anti-HMGB1 (1:500, rabbit polyclonal; Abcam Cat#ab18256, RRID:AB_444360) overnight at 4˚C, washed with TBS-T, then incubated for 1 hr with IRDye 680RD donkey anti-rabbit IgG H+L (1:10,000, Millenium Science/Li-Cor). Semi-quantification of the bands was performed with a LI-COR Odyssey scanner and software (LI-COR Biosciences).

## Airway function assessment

Airways resistance was measured by forced oscillation technique (Flexivent, Scireq), as described previously [37].

## ASM culture

Tracheal ASM cells were isolated from neonatal WT mice and cultured in DMEM/F-12 with 100 units/ml penicillin, 100 μg/ml streptomycin, 0.25 μg/ml amphotericin B, and 10% FBS as described [52]. Cell purity, as assessed by alpha-smooth muscle actin staining, was greater than 98%. Experiments were performed at passage 4. $5 \times 10^4$ Cells were loaded with violet proliferation dye 450 according to the manufacturer's instructions (BD Biosciences), and proliferation indicated by decreasing median fluorescence intensity (MFI) was measured by flow cytometry at day 4. ASM cells were stimulated with IL-13, IL-33 (both R&D Systems), disulphide HMGB1 or non-oxidising HMGB1 (both HMGBiotech). For cytokine or receptor blockade, cells were treated with sIL-13Rα2-Fc (5 μg/ml; Pfizer, Inc), LPS-RS (1 μg/ml, Invivogen), anti-HMGB1 (clone 2G7; 5 μg/ml) or FPS-ZM-1 (300 nM; Calbiochem). Supernatants were stored at -80ºC until analysis.

## ILC2s culture and activation

At 10 dpi, the lungs of IRF7$^{-/-}$ or RAGE$^{-/-}$ mice were excised and digested using the gentleMACS dissociation kit (Miltenyi Biotech). Natural ILC2s [Lineage$^-$ (CD45R, CD3, CD4, CD11c, CD19, Gr-1, CD11b, CD2, NK1.1, CD49b), CD90.2$^+$, CD45$^+$, CD25$^+$, ST2$^+$, Sca-1$^+$] were FACS-sorted (>98% purity) and confirmed in separate studies that these ILC2s are GATA3+. ILC2s were cultured at $10^4$ cells/well in 200 μL of complete medium (RPMI-1640 with 1 mM sodium pyruvate, 10% FBS, 2 mM L-gutamine, 20 mM HEPES, 100 units/ml penicillin-streptomycin, 50 μM 2-mercaptoethanol) and stimulated with the indicated combinations of IL-2 (30 ng/ml; ebioscience) and IL-33 (3 ng/ml) for 4 days. To determine HMGB1-induced ILC2 cytokines production, ILC2s were cultured in IL-2 +/- disulphide-HMGB1 (30 ng/ml) or non-oxidisable mutant HMGB1 (30 ng/ml) as indicated. For cytokine or receptor blockade, cells were treated with soluble IL-13Rα2-Fc (5 μg/ml; Pfizer Inc), TLR4 antagonist, LPS-RS (1 μg/ml; Invivogen); RAGE antagonist, FPS-ZM-1 (300 nM; Calbiochem) or anti-HMGB1 (clone 2G7, 5 μg/ml; Dr.

Tracey, The Feinstein Institute for Medical Research). ILC2 proliferation and cell death was measured by CFSE (Sigma Aldrich) and annexin V (Biolegend) respectively.

## Statistical analysis

Graphical and statistical analyses were generated with GraphPad Prism (La Jolla, California). A Student's t-test, one-way ANOVA with a Dunnett post-hoc test, or two-way ANOVA with a Tukey post-hoc test was applied as appropriate. A $P$ value <0.05 was considered statistically significant.

## Supporting information

**S1 Fig. PVM infection induces severe bronchiolitis, type-2 inflammation and ASM remodelling in IRF7 deficient mice.** WT (IRF7$^{+/+}$, closed circle) and IRF7$^{-/-}$ mice (open circle) were inoculated with PVM or vehicle at postnatal day 7 and samples collected at 3, 7, 10 and 14 days post infection (dpi). (a) IL-12p40 protein expression in BALF. (b) IL-33 protein expression in lungs. (c) Eosinophils in BALF. (d) Gating strategy for ILC2s: Lineage$^-$ (CD45R, CD3, CD4, CD11c, CD19, Gr-1, CD11b, CD2, NK1.1, CD49b), CD90.2$^+$, CD45$^+$, CD25$^+$, ST2$^+$, Sca-1$^+$ ILC2 cells. (e) Numbers of ILC2 cells in left lung lobe and BALF in WT and IRF7$^{-/-}$ mice at 10 dpi. (f) Representative micrograph (x1000 magnification) of PCNA immunoreactivity (red) and smooth muscle actin immunoreactivity (green) in the lung at 10 dpi (left), quantified as % of ASM cells (right). Scale bar = 5 μm. Data are representative of $n$ = 2 experiments with 3 to 8 mice in each group and are presented as mean ± SEM (a-c) or scatter plot (e) or as box-and-whisker plots showing quartiles (boxes) and range (whiskers; f). Data were analysed by Two-way ANOVA with Tukey post hoc test (a-d) or one-way ANOVA with Dunnett post hoc test (f) or T-test (g); $^*$, P < 0.05; $^{**}$, P < 0.01; $^{***}$, P < 0.001 compared with the WT control group. (TIF)

**S2 Fig. Secondary virus infection induces an asthma-like pathology: ASM remodelling, basement membrane thickening and airway hyperreactivity.** (a) Six weeks after the primary infection, WT (IRF7$^{+/+}$) and IRF7$^{-/-}$ mice were re-infected with PVM and ASM area assessed 8 weeks later. (b) Subepithelial collagen deposition and (c) AHR at 0, 3, 7 and 21 days post secondary infection. (d) Study design. WT (IRF7$^{+/+}$) and IRF7$^{-/-}$ mice were either inoculated with PVM at postnatal day 7 or exposed to cockroach allergen (CRE) 6 weeks later or both. (e) Total ILC2 (Lineage$^-$ (CD45R, CD3, CD4, CD11c, CD19, Gr-1, CD11b, CD2, NK1.1, CD49b), CD90.2$^+$CD45$^+$CD25$^+$ST2$^+$Sca-1$^+$), eosinophil (Siglec F$^+$CD11b$^+$Ly6G$^{int}$MHCII$^-$CD3$^-$B220$^-$) and CD4$^+$ T cells (CD3$^+$ CD4$^+$ CD8$^-$) numbers in BAL were identified by flow cytometry, followed by mucus-producing airway epithelial cells (AECs) and ASM area quantification. Data are representative of $n$ = 2 experiments with 4 to 8 mice in each group and are presented as box-and-whisker plots showing quartiles (boxes) and range (whiskers; a, e) or as mean ± SEM (b-c). Data were analysed by T-test (a) or Two-way ANOVA with Tukey post hoc test (b-c) or one-way ANOVA with Dunnett post hoc test (e). $^*$, P < 0.05; $^{**}$, P < 0.01; $^{***}$, P < 0.001 compared with the WT control group. (TIF)

**S3 Fig. Effect of various interventions on type-2 inflammation and ASM remodelling.** (a) Study design. (b) IL-5 expression in IRF7$^{-/-}$ mice in BALF (left) and eosinophils in BALF (right). (c-i) Effect of glycyrrhizin on type-2 inflammation in PVM-infected IRF7$^{-/-}$ mice. (c) Study design. (d) IL-33, (e) IL-5 and (f) IL-13 protein expression in IRF7$^-$/$^-$ mice in BALF at 10 dpi. (g) ILC2s in lung. (h) Eosinophils in BALF. (i) ASM area. Data are representative of $n$ = 2 experiments with four to eight neonates in each group and are presented as box-and-whisker

plots showing quartiles (boxes) and range (whiskers; b, d-i). Data were analysed by T-test; *, P < 0.05; **, P < 0.01; ***, P < 0.001 compared with the IRF7$^{-/-}$ control group.
(TIF)

**S4 Fig. IL-13 induces ASM proliferation.** (a) Dose response curve of IL-13 or IL-33 stimulated ASM proliferation. (b) ILC2 purification strategy by FACS; Lineage$^-$ gate included CD45R, CD3, CD4, CD11c, CD19, Gr-1, CD11b, CD2, NK1.1, CD49b. The sorted cells were positive for the following four markers: CD90.2$^+$, CD45$^+$, CD25$^+$, and ST2$^+$. (c) IL-2 +/- IL-33 stimulated IL-5, IL-9 IL-13, IFN-γ, IL-4 and IL-17A protein production by ILC2s. (d) ILC2s numbers (left panel), IL-5 (middle panel) and IL-13 (right panel) protein production by lung ILC2s purified from WT or IRF7$^{-/-}$ mice cultured with IL-2 +/- IL-33 (30 ng/ml) for four days. (e) ASM proliferation in response to IL-13 in the presence of LPS-RS (TLR4 antagonist; left) or FPS-ZM1 (RAGE antagonist; right). (f) Percent dead (left) or proliferating ILC2 cells in response to anti-HMGB1 (2G7), TLR4 antagonist (LPS-RS) or RAGE antagonist (FPS-ZM1; right). Data are representative of *one* experiment, performed twice with similar results. Data is shown as mean ± SEM (a) or as box-and-whisker plots showing quartiles (boxes) and range (whiskers; c-f). Data were analysed by Two-way ANOVA with Tukey post hoc test (a) or T test ((c) or One-way ANOVA with Dunnett post hoc test (d-f); *, P < 0.05; **, P < 0.01; ***, P < 0.001 compared with the respective control group.
(TIF)

## Author Contributions

**Conceptualization:** Zhixuan Loh, Simon Phipps.

**Data curation:** Zhixuan Loh, Jennifer Simpson, Ashik Ullah, Vivian Zhang, Choon Boon Sim, Enzo Porrello, Simon Phipps.

**Formal analysis:** Zhixuan Loh, Jennifer Simpson, Katie Lane, Choon Boon Sim, Enzo Porrello, Raymond J. Steptoe, Kirsten M. Spann, Maria B. Sukkar, Simon Phipps.

**Funding acquisition:** Simon Phipps.

**Investigation:** Zhixuan Loh, Jennifer Simpson, Ashik Ullah, Choon Boon Sim, Enzo Porrello, Kirsten M. Spann, Maria B. Sukkar, Simon Phipps.

**Methodology:** Zhixuan Loh, Simon Phipps.

**Supervision:** Simon Phipps.

**Validation:** Zhixuan Loh, Jennifer Simpson, Ashik Ullah, Vivian Zhang, Katie Lane, Choon Boon Sim, Enzo Porrello, Raymond J. Steptoe, Kirsten M. Spann, Maria B. Sukkar, Simon Phipps.

**Writing – original draft:** Zhixuan Loh, Jennifer Simpson, Simon Phipps.

**Writing – review & editing:** Zhixuan Loh, Jennifer Simpson, Ashik Ullah, Vivian Zhang, Wan J. Gan, Jason P. Lynch, Rhiannon B. Werder, Al Amin Sikder, Katie Lane, Choon Boon Sim, Enzo Porrello, Stuart B. Mazzone, Peter D. Sly, Raymond J. Steptoe, Kirsten M. Spann, Maria B. Sukkar, John W. Upham, Simon Phipps.

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
