## [Decision Letter · Decision Letter 0]

26 Apr 2020

Dear A/Prof Phipps,

Thank you very much for submitting your manuscript "HMGB1 amplifies ILC2-induced type-2 inflammation and airway smooth muscle remodelling." for consideration at PLOS Pathogens. As with all papers reviewed by the journal, your manuscript was reviewed by members of the editorial board and by several independent reviewers. The reviewers appreciated the attention to an important topic. Based on the reviews, we are likely to accept this manuscript for publication, providing that you modify the manuscript according to the review recommendations.

Sincerely,

Mark T. Heise

Section Editor

PLOS Pathogens

Kasturi Haldar

Editor-in-Chief

PLOS Pathogens

orcid.org/0000-0001-5065-158X

Michael Malim

Editor-in-Chief

PLOS Pathogens

orcid.org/0000-0002-7699-2064

Reviewer Comments (if any, and for reference):

Reviewer's Responses to Questions

**Part I - Summary**

Reviewer #1: This manuscript by Loh et al utilizes pneumonia virus of mouse (PVM) infection in IRF7-deficient mice, which have impaired anti-viral immunity due to an impaired IFN-alpha response, to understand the role of ILC2 and HMGB1 in viral-induced asthma. A major strength of the paper is the use of a clinically relevant model. PMV replicates several of the features of severe human RSV bronchiolitis, and neonatal animals are used. Furthermore, differential expression of IRF7 may be linked to asthma.

The major findings include:

1. Impaired anti-viral immunity in IRF7-deficient mice is associated with increased epithelial cell injury, increased IL-33 and HMGB1, increased lung ILC2 (the major source of IL-13) and type 2 inflammation, and increased in airway smooth muscle (ASM). Type 2 inflammation is also observed after re-infecting IRF7-deficient mice with PVM and is associated with increased ASM and increased airways hyper-responsiveness (which is modest at best and only observed at the highest methacholine dose). Treatment of IRF7-deficient mice with anti-HMGB1 or anti-IL-33 abrogates lung ILC2 numbers, type 2 inflammation and ASM growth following PVM infection. These findings are important but not surprising.

2. In vitro, IL-13 production by IL-33 treated ILC2 promotes ASM proliferation and ASM expression of TGF-beta and HMGB1. Furthermore, IL-13 acts in an autocrine fashion to increase ILC2 cytoplasmic HMGB1 and release, and HMGB1 in turn stimulates ILC2 (via its receptor RAGE) to produce type 2 cytokines. Treatment of PVM-infected IRF7-deficient mice with a RAGE antagonist decreased type 2 inflammation. Of note, HMBG1-mediated activation of ILC2 has recently been demonstrated in using a mouse model of allergic airway inflammation.

Overall, the manuscript is well written, the results are clear and the topic is of significant interest.

Reviewer #2: In the current study Phipps and colleagues investigate in the role of HMGB1 in augmenting ILC2 dependent type-2 inflammation in a mouse model of severe respiratory viral infection in early life. The authors that HMGB1 promotes type 2 immunity in a mouse model of severe viral pneumonia, highlighting the promotion of airway smooth muscle. They find this is associated with increases in ILC2, but not Th2, numbers and ILC2s can directly signal on ASM cells to promote their proliferation through a HMGB1-IL-13-TGFb axis and involves HMGB1’s ability to signal through both TLR4 and RAGE. The work builds on previous studies by the group, and in particular, is closely linked to a recent study in the AJRCCM, in which type-2 enhanced inflammation as a result of HMGB1/IRF7 during PVM infection is also described, but the role of epithelial cell death as a mechanism explored (Simpson et al, 2020, PMID 32105156). The two studies therefore do explore distinct experimental mechanisms and the current study provide new insights into the potential mechanisms underlying long term effects of early life viral bronchiolitis with regards to airway remodelling. Overall the study well described and carried out, and the mechanisms underlying viral-induced wheeze and asthma are of both clinical importance and biological interest. There are however a number of areas where additional information or clarification would be beneficial.

**Part II – Major Issues: Key Experiments Required for Acceptance**

Reviewer #1: 1. In many of the figures, some of the data points are missing error bars. Is this because the errors are smaller than the symbols used? Or is it because of the number of animals analyzed at a given time point? In other words, when 4-8 animals were used per group, does this mean that 4-8 animals were analyzed at each time point or 4-8 animals for the entire experiment with only 1-2 animals per time point?

2. Fig 2: In panel (c), no cells are included in the CD25+CD90.2+ ILC2 gate, yet the quantification of IL-13 positive cells on the subsequent panel is 6%. Is this truly representative? It is difficult to assess whether (a) ILC2 in WT are not producing IL-13, (b) there are just fewer numbers of ILC2 in WT mice or (c) there are reductions in both the number of ILC2 and the production of IL-13 by ILC2 in WT mice.

Reviewer #2: 1. One seemingly important question arsing from the authors work is whether the effects they see on ASM are the result of enhanced type 2 immunity due to IRF7 deficiency are dependent or independent heightened viral loads and damage (such as necroptosis they have previously reported). Can the authors show the effect of a non-infectious stimuli on ILC2/ASM in IRF7-/- vs WT mice. For instance TLR7 is a potent stimulator of ISGs while intranasal rIL-33 potently promotes ILC2 responses and AHR. This would seem particularly important in the context of segregating the role of HMGB1 on epithelial cells and the effect of ILC2s on ASM.

2. The authors analysis of long term consequences of early life PVM infection are interesting but lack depth. Early life infection with hRSV for instance has been shown to prime Th2 cells capable of eliciting type-2 enhanced disease on either homologous or heterologous rechallenge (e.g. Culley et al, J Exp Med, 2002, PMID 12438429). In the PVM and CRE re-challenge models, can the authors show the relative contribution of ILC2s versus other T cells to this effect? Were the number of each enumerated? In addition, can the authors clarify the why re-challenge with an apparently low dose of PVM fails to elicit infection but promotes prominent inflammation?

3. TGF-beta is clearly a critical downstream mediator in the authors proposed model, however there seems to be an incomplete assessment of its sources in vivo and activation in vivo. For instance the authors focus on ASM as the relevant source of TGFb however their histology in figure 2F suggests prominent TGFb staining in IRF7-/- mice? This could be particular relevant given that epithelial derived TGF-beta has previously been shown to augment type-2 lung inflammation through increased recruitment of ILC2s (Denney et al, Immunity, 2015, PMID 26588780). Can the authors measured active TGFb in the airways of IRF7-/- versus WT PVM infected mice? Also, although minor, can the authors also clarify precisely which ELISA they used to identify active, versus latent, TGFb as traditionally this has been challenging via ELISA, and assessment has relied on a bioassays for downstream signalling? Finally in their co-culture system can the authors speculate which cell type is responsible for the activation of TGF-b – do either ILC2s or ASMs express potential activating molecules such as �v �6/8, MMP2/9, plasmin.

**Part III – Minor Issues: Editorial and Data Presentation Modifications**

Reviewer #1: 1. Fig. 1: Given the significant differences in neutrophilic infiltrate, airway epithelial cells sloughing and edema, it would be useful to see representative histology.

2. Fig. 1: The time point assessed in panels (h) and (i) should be clarified in the figure or the legend. Additionally, it would be helpful to use the closed vs open circle designation for WT vs IRF7-deficient mice in panel (i) (as is used in the rest of this figure).

3. Fig. 1 and S1: The difference in IL-33 in the lung between WT and IRF7-deficient mice following PVM infection (Fig. S1) is more impressive than the differences in the BAL (Fig. 1). This is not surprising as IL-33 is released basolaterally from AEC. I assume BAL fluid is shown in Fig. 1 because only BAL (and not lung) levels are available for the experiments in Fig. 3, 4 and 6?

4. Fig. S1: In the legend, panel (e) is described as demonstrating IL-4 producing ILC2, CD4 and CD8 T cells. However, the results section describes this panel only as a quantification of ILC2 in the lung as compared to BAL fluid, without mentioning IL-4 expression, and CD4 and CD8 T cells are not displayed.

5. Fig. 2: If the data are available, it would be helpful to see quantification of IL-5 at time points other than day 10.

6. Fig. 5: It is not clear that panel (a) adds anything to the data shown in panel (b).

7. Fig. 5: In panel (e), the cytoplasmic HMGB1 staining show is rather subtle. It would be helpful to see a negative control or WT mouse.

Reviewer #2: 1. The authors should remove all data pertaining to anti-CD25 depletion (Figure S4). While it is understandable that the authors sought to carry out an in vivo experiment evaluating the role of ILC2s in ASM remodelling, and the complexity of ILC2 specific deficient models presents some technical difficulties, the routine use of the same anti-CD25 regime to deplete Tregs (in spite of the surprising failure to have an effect in the lungs in this study) in the majority of other in vivo experiments raises more questions that it resolves. It is also unnecessary for the conclusions drawn from this study.

2. Figure 1i – can the authors clarify in the figure legend whether the western blot was carried out with or without infection (and at what d.p.i.). If with infection, can the authors confirm IRF7-/- mice have no baseline alteration in the presence of HMGB1 isoforms without infection?

3. Line 159 – remove the work “remarkably”.

4. Line 222 – it would be helpful if the authors could include a line describing the known proliferative effect of FBS on ASM cells in vitro, the mechanism behind this, and its use as a known positive control in this set of experiments.

5. Fig 2d – the empty and filled symbols are swapped compared to the rest of the figure. (WT empty/KO filled).

6. Fig 1a. Use of % of AEC stained histologically positive for PVM seems an unconventional measure of viral replication, whole lung analysis using either RT-qPCR or especially traditional plaque assay/TCID50 would seem more appropriate (and account for replication anywhere in the lungs, not only in the airway epithelium).

PLOS authors have the option to publish the peer review history of their article (what does this mean?). If published, this will include your full peer review and any attached files.

Reviewer #1: Yes: Josalyn L. Cho

Reviewer #2: No
---

## [Editor Report · Decision Letter 1]

24 May 2020

Dear A/Prof Phipps,

We are pleased to inform you that your manuscript 'HMGB1 amplifies ILC2-induced type-2 inflammation and airway smooth muscle remodelling.' has been provisionally accepted for publication in PLOS Pathogens.

Best regards,

Mark T. Heise

Section Editor

PLOS Pathogens

Mark Heise

Section Editor

PLOS Pathogens

Kasturi Haldar

Editor-in-Chief

PLOS Pathogens

orcid.org/0000-0001-5065-158X

Michael Malim

Editor-in-Chief

PLOS Pathogens

orcid.org/0000-0002-7699-2064
---

## [Editor Report · Acceptance letter]

23 Jun 2020

Dear A/Prof Phipps,

We are delighted to inform you that your manuscript, "HMGB1 amplifies ILC2-induced type-2 inflammation and airway smooth muscle remodelling," has been formally accepted for publication in PLOS Pathogens.

Best regards,

Kasturi Haldar

Editor-in-Chief

PLOS Pathogens

orcid.org/0000-0001-5065-158X

Michael Malim

Editor-in-Chief

PLOS Pathogens

orcid.org/0000-0002-7699-2064